# Iron Trace Elements Concentration in PM_10_ and Alzheimer’s Disease in Lima, Peru: Ecological Study

**DOI:** 10.3390/biomedicines12092043

**Published:** 2024-09-08

**Authors:** Diego Fano-Sizgorich, Cinthya Vásquez-Velásquez, Carol Ordoñez-Aquino, Odón Sánchez-Ccoyllo, Vilma Tapia, Gustavo F. Gonzales

**Affiliations:** 1Laboratorio de Endocrinología y Reproducción, Facultad de Ciencias e Ingeniería, Universidad Peruana Cayetano Heredia, Lima 15102, Peru; diego.fano.s@upch.pe (D.F.-S.); carol.ordonez.a@upch.pe (C.O.-A.); vilma.tapia.a@upch.pe (V.T.); gustavo.gonzales@upch.pe (G.F.G.); 2Departamento de Ingeniería, Facultad de Ciencias e Ingeniería, Universidad Peruana Cayetano Heredia, Lima 15102, Peru; 3Grupo de Investigación en Contaminación Atmosférica, Facultad de Ingeniería y Gestión, Universidad Nacional Tecnológica de Lima Sur, Lima 15834, Peru; osanchezbr@gmail.com

**Keywords:** iron, Alzheimer’s disease, particulate matter, air pollution, Latin America

## Abstract

Alzheimer’s disease (AD) has been linked to air pollution, especially particulate matter (PM). PM comprises various elements, including iron-rich particles that may reach the brain through inhalation. Lima, Peru is one of the most polluted cities in Latin America, with a high rate of AD. The study aims to evaluate the association between iron (Fe) trace elements in PM_10_ and AD cases in Lima, Peru. This retrospective ecological study used monthly Fe concentration data from the Peruvian Ministry of Health. AD cases (ICD-10-G30) and dementia in AD cases (DAD, ICD-10-F00) were obtained from the Peruvian CDC. Fe trace element data were available for six districts in Lima for the years 2017–2019 and 2022. Cases were standardized based on ≥60-year-old populations of each district. Hierarchical mixed-effects models of Gaussian and negative binomial families were constructed to evaluate both outcomes jointly (AD + DAD) and separately (AD, and DAD). A sensitivity analysis was conducted by excluding data from Lima’s downtown district. In the complete model, log-Fe concentration was associated with a higher rate of AD + DAD and DAD, and with a higher IRR for the three outcomes. After controlling for other metals, a higher DAD rate was observed (β-coeff = 6.76, 95%CI 0.07; 13.46, *p* = 0.048), and a higher IRR for AD + DAD (1.55, 95%CI 1.09; 2.20, *p* = 0.014) and DAD (1.83, 95%CI 1.21; 2.78, *p* = 0.004). The association was not significant in the sensitivity analysis. In conclusion, exposure to Fe through PM_10_ inhalation may be associated with the presence of AD in Lima.

## 1. Introduction

Alzheimer’s disease (AD) is a neurodegenerative condition characterized by the progressive loss of memory, cognitive functions, and learning abilities [1]. AD is a pathology characterized by the progressive loss of neuronal connections, which leads to gray matter atrophy in the brain [2]. The global burden of AD has steadily increased over the past few decades [3]. The 2024 report from the Alzheimer’s Association estimated that 6.9 million adults >65 years in the US are currently living with AD. Modeling studies predict that this number will double by 2050.

The cause of Alzheimer’s disease is still not completely clear. There are genetic and environmental risk factors [4]. It is most likely that the cause is not a single factor, but rather a complex interaction of external influences and internal changes occurring in the body. Different significant modifiable risk factors for AD, such as air pollution, have been identified [5].

Air pollution, seen as particulate matter (PM), may cause oxidative stress and neuroinflammation and contribute to the deposition of amyloid beta (Aβ) in the brain [6]. Particulate matter has been observed in olfactory bulb neurons. Exposure to air pollution causes neuroinflammation and alters brain innate immune response, promoting the accumulation of Aβ42 and alpha-synuclein even during childhood [7]. 

One-year PM_2.5_ exposure has been associated with decreased Aβ42 in cerebrospinal fluid (CSF), suggesting an accumulation of amyloid plaques in the brain and an increased risk of developing AD [8]. Notably, exposure to PM under 10 µm (PM_10_) has been identified as a potential contributor to AD [9,10,11,12,13].

PM consists of diverse particles, including coal, organic compounds, and various trace metals [14]. Trace elements such as iron (Fe), zinc (Zn), copper (Cu), and manganese (Mn) are absorbed from food via the gastrointestinal tract, transported into the brain, and play central roles in normal brain functions. Oxidative stress resulting from abnormal homeostasis of transition metals such as Fe, Cu, and Zn may contribute to AD [15].

High levels of Fe, Cu, and Mn are associated with mild cognitive impairment (MCI) and AD, while low selenium (Se) levels are linked to poor cognitive status [16,17,18]. High aluminum (Al) exposure is associated with frontotemporal dementia (FTD), and elevated Se levels may be linked to its onset. Se, in conjunction with Fe, plays a distinct role in the process of ferroptosis [16,17].

Abnormal Fe concentrations in different brain regions of AD patients have been reported, and associated with cognitive impairment due to local inflammation, affecting nerve function [19], stimulating amyloid aggregates formed from the β-amyloid peptide [20], and increasing the production of reactive oxygen species (ROS), contributing to the pathogenicity of AD [21]. 

In AD, a diffuse accumulation of iron occurs in various regions, such as the cortex and hippocampus [22]. Iron accumulates in regions affected by AD as the brain ages [23]. Moreover, growing evidence sustains the significant impact of Fe metabolism in relation to other pathological processes encountered in the AD-affected brain, such as the amyloidogenic pathway, chronic inflammation, or oxidative stress-inducing neuronal vulnerability [8,24,25]. 

It is hypothesized that iron also could reach brain tissue via the olfactory bulb during PM inhalation [24,26]. This process may lead to a gradual deposition, as evidenced by the presence of nanoparticles of magnetite in human brain samples [27]. 

Ferroptosis, an iron-dependent programmed cell death, has been implicated in the pathological changes associated with AD. Iron is known to influence Tau phosphorylation, resulting in excessive phosphorylation and the promotion of neurofibrillary tangles (NFTs) [28].

SO_2_ emissions, which lead to high concentrations of SO_2_ in the air, generally also lead to the formation of other sulfur oxides (SO_x_). SO_x_ can react with other compounds in the atmosphere to form small particles. These particles contribute to PM pollution. SO_2_ facilitates Fe uptake through the bronchial epithelium and alters its intracellular distribution [29]. Thus, a synergy between Fe and SO_2_ pollutants in other organs like the brain is possible. 

The association of air pollution on AD should be considered [30,31]. Lima, the capital of Peru, is one of the most polluted megacities in Latin America, and it is important to know if iron in PM_10_ is associated with AD. Exposure to ambient fine particles and gaseous pollutants such as SO_2_ significantly increased the accumulation of Aβ42 in both male and female rats after three months [32]. This study aims to evaluate the association between Fe concentration in PM_10_ and AD cases in Lima.

## 2. Materials and Methods

### 2.1. Study Design and Study Area

This study employed a retrospective ecological design, in which monthly concentrations of Fe trace elements in PM_10_ and 13 other trace metals, and the number of AD cases (ICD-10 code: G30) and dementia in AD (DAD) cases (ICD-10 code: F00) were obtained for six districts in Lima. 

The Metropolitan Area of Lima, Peru’s capital city, comprises 43 districts with a population exceeding 10 million. The districts covered in this study were Comas (north), San Juan de Miraflores and Santiago de Surco (south), El Agustino (east), Lima Downtown, and Lince (center). All these districts, characterized by bustling avenues, have undergone significant population and commercial growth in the last decades.

### 2.2. Data Collection

Monthly concentrations of PM_10_ (µg/m^3^) and Fe (ng/m^3^) in PM_10_ for the six Lima districts were obtained from the DIGESA (Dirección General de Salud Ambiental e Inocuidad Alimentaria, in Spanish) website. According to DIGESA, PM_10_ was sampled using a “PM_10_ high volume air sampler” that collected ambient PM with an aerodynamic diameter of 10 µm or less. 

Ambient air PM_10_ samples were collected on quartz filters weekly. These samples underwent chemical analysis using inductively coupled plasma–mass spectrometry (ICP-MS) [33] to determine the concentrations of various heavy metals. DIGESA performs an extensive speciation process of 22 metals. The elements Al, Ba, Ca, P, K, Mg, Sr, and Ti were not considered due to poor data availability. Therefore, from 5 January 2017 to 26 December 2019, the following 14 heavy metals were analyzed: Fe, Be, Cd, Co, Cr, Cu, Li, Mn, Mo, Ni, Pb, Sb, Se, and Zn. These data were sourced from DIGESA’s website. This manuscript primarily focuses on the analysis of Fe concentrations in PM_10_. 

We also downloaded the data of SO_2_ concentrations for the same period. According to DIGESA, ambient SO_2_ concentrations in Lima city were measured using the Fluorescence SO_2_ Analyzer. This analyzer operates on the principle that SO_2_ molecules absorb ultraviolet light. Upon absorbing this light, the SO_2_ molecules become excited and subsequently emit light as they return to their ground state. The intensity of the emitted light is measured, and this intensity is directly proportional to the concentration of SO_2_ in the air. 

The monthly case numbers of AD (AD, ICD-code G30) and dementia in AD (DAD, ICD-code F00) were acquired from the Peruvian Center for Disease Control and Prevention of the Ministry of Health for 2017 to 2019 and 2022. The period from 2020 to 2021 was not considered due to a potential underdiagnosis during the COVID-19 pandemic, as observed in Spain [34]. We opted for these two ICD codes as they are both specific to Alzheimer’s disease and may represent different individuals or cases. Previous studies have employed these codes to define their AD outcomes [35,36]. The ≥60-year-old population in each district for these periods was retrieved from the Single National Health Information Repository (REUNIS) of the Ministry of Health. This age group is considered to be at the highest AD risk.

### 2.3. Statistical Analysis

For each year, the numbers of AD, DAD, and AD + DAD cases were standardized using the district population for that specific year. Fe concentrations in PM_10_ were log-transformed to achieve a normal distribution. Additionally, Pearson correlation analysis was performed between Fe concentration and the other different metals, as well as with the different AD outcomes (AD, DAD, and AD + DAD). Correlation analysis was weighted by the district population.

To assess mean log-Fe concentration differences between districts, a one-way analysis of variance and post hoc Bonferroni test were employed. Normal distribution of log-Fe concentration in each district was evaluated using Q-Q plots. The homoscedasticity assumption was evaluated using Bartlett’s test. The ANOVA test with the post hoc test was used to compare mean disease incidences between districts.

Two sets of hierarchical models were employed to analyze the relationship between iron concentrations and the number of AD cases while accounting for the hierarchical structure of the data (observations nested within districts). A linear mixed-effects regression model was used to further explore the relationship between the logarithm of iron concentration with the rate of AD outcomes (AD, DAD, and AD + DAD). The association measure calculated was the β-coefficient. A mixed-effects negative binomial regression model was then employed to account for overdispersion of the number of cases, considering the district population for different years (2027, 2018, 2019, and 2022) as an offset. The association measure calculated was the incidence rate ratio (IRR). Both hierarchical models included month and year as fixed effects.

Afterwards, considering that other metals could be also associated with AD cases, the models were adjusted for correlated metal (Pb, Cu, Zn, and Mn). The mixed-effects models were evaluated using likelihood ratio tests to compare them against simpler models without random effects. The significance of the fixed effects was assessed using Wald chi-square tests. A secondary analysis including log-Fe and SO_2_ concentration was performed; for this, a generalized linear model (GLM) of Gaussian and negative binomial families were considered. The GLM approach was preferred over hierarchical models due to the low number of observations for SO_2_ (n = 78).

A sensitivity analysis was conducted by excluding data from Lima Downtown due to the presence of the Instituto Nacional de Ciencias Neurológicas (INCN). This institution is the largest neurology center in the country, and its inclusion might introduce selection bias due to its ongoing prevention and diagnostic campaigns. This analysis was repeated for each ICD-10 outcome separately (AD, DAD, and AD + DAD). The sensitivity analysis was also performed for the correlation analysis.

The statistical software used was STATA version 17 (StataCorp, College Station, TX, USA, RRID: SCR_012763). Significance was considered when *p* < 0.05.

## 3. Results

The mean Fe concentration in PM_10_ for Lima was 1160 ± 660 ng/m^3^. Notably, as seen in Table 1, districts like Comas, San Juan de Miraflores, El Agustino, and Lima Downtown exhibited the highest Fe levels in PM_10_, while Lince and Santiago de Surco had the lowest concentrations (*p* < 0.001). A significant difference in mean Fe concentration was observed between districts (ANOVA test *p* < 0.001). Appendix A shows that the trend of iron in the Comas district shows high Fe concentration values (4346 ng/m^3^) during the periods of March (summer) and April (autumn), and low Fe concentration values of 853 ng/m^3^ in July (winter) and September (spring). In the districts of Lima Downtown, Lince, and El Agustino, Fe increased from 2017 until April 2018, November 2018, and March 2018, respectively, followed by a slight decrease in Fe concentration until 2022. Meanwhile, in the district of San Juan de Miraflores, there is a trend of high daily Fe concentration values in April (3050 ng/m^3^) and low values in July and September of 398 ng/m^3^. In the district of Santiago de Surco, starting from October 2017 (1196 ng/m^3^), a slight decrease in Fe concentration is noted until December 2022, with a value of 606 ng/m^3^.

From the 14 metals evaluated in PM_10_, only 6 showed differences between districts (Fe, Cu, Mn, Ni, Pb, and Zn). For Fe, Mn, Ni, and Pb, the highest levels were observed in Comas, while the lowest levels were in Lince and Santiago de Surco. The highest levels in San Juan de Miraflores were observed for Fe and Cu. The box plots of Fe and other metals can be seen in Appendix A, which shows that the districts of Comas and San Juan de Miraflores typically present the highest levels of metals, although with greater dispersion compared to the other districts.

Figure 1A shows the correlation analysis results between the different trace metals in PM_10_ and AD outcomes. Fe had a significant positive correlation with Ni, Sb, Pb, Zn, Cu, Mn, and Se (*p* < 0.001); and with Be, Li, and Cr (*p* < 0.05). The highest correlation of iron was observed with Zn (r = 0.795) and Pb (r = 0.786). AD + DAD had a significant negative correlation with Li, Pb, and Cu (*p* = 0.003), Be, Zn and Se (*p* < 0.05). AD showed a negative correlation with Li, Pb, Zn, Cu, Se, and Fe (*p* < 0.002), and Be and Ni (*p* < 0.05). DAD only showed a negative correlation with Li (*p* = 0.034), Cu (*p* < 0.001), and Mn (*p* = 0.046).

Figure 1B shows that the correlation between the outcomes and metals was different in the sensitivity analysis (without Lima Downtown). AD + DAD had a significant negative correlation with Li and Se (*p* < 0.05), and a positive correlation with Mn and Fe (*p* < 0.001). AD showed a negative correlation with Ni and Se (*p* < 0.003), and a positive correlation with Mo (*p* = 0.014) and Fe (*p* < 0.001). DAD only had a positive significant correlation with Fe (*p* = 0.017) and Mn (*p* = 0.002), and a negative correlation with Cu (*p* = 0.005) and Li (*p* = 0.041).

In Appendix A, the correlation analysis between Fe and the other metals by district is shown. No positive correlation was observed between Fe, Mo, and Sb in any of the six districts studied. A positive correlation was observed between Fe and Be, Cd, Co, Li, and Se in Comas and San Juan de Miraflores. A positive correlation was observed between Fe and Ni in Comas and Lima downtown, but a negative correlation was observed in Santiago de Surco. A positive correlation between Fe and Pb and between Fe and Zn was observed in five districts, with the exception being Santiago de Surco. Cu was correlated with Fe in four districts, with the exceptions being San Juan de Miraflores and Santiago de Surco. Fe was positively correlated with Cr in Lima Downtown and El Agustino. Fe was positively correlated with Mn in Comas, El Agustino, and San Juan de Miraflores.

Lima Downtown accounted for most of the AD (71.18%) and DAD cases (34.98%). AD + DAD incidences were significantly higher in Lima Downtown compared to other districts (*p* < 0.001). However, Lima Downtown and El Agustino had the highest incidence of DAD (*p* < 0.001) (Table 2). The lowest Fe levels in PM_10_ were found in Lince and Santiago de Surco, with no significant differences between them (Bonferroni post hoc *p* > 0.050) (Table 1). These districts also reported the lowest incidence of AD + DAD (4.50 and 0.89 cases per 100,000). Similar patterns were observed for AD and DAD separately (Table 2).

In the scatter plots showing the rates of AD, DAD, and AD + DAD per 100,000 people versus Fe concentration, a non-significant negative trend was observed for AD (r = −0.106, *p* = 0.098) and AD + DAD (r = −0.062, *p* = 0.338), and for DAD a null correlation was found (r = −0.002, *p* = 0.971) (Figure 2), in which most of the cases were observed around an iron concentration of 1000 ng/m^3^. After excluding Lima Downtown data, the negative trend was reversed, although the correlation was still non-significant for AD (r = 0.098, *p* = 0.166), DAD (r = 0.027, *p* = 0.708), and AD + DAD (r = 0.058, *p* = 0.411). This suggested a need to perform a sensitivity analysis excluding Lima Downtown for the regression models.

In the assessed mixed-effects models, the full model (including all districts) showed a significant association between an increase in log-Fe concentration and AD + DAD for both Gaussian and negative binomial models, increasing the AD + DAD rate per 100,000 people in 9.93 (95%CI 2.93; 16.93, *p* = 0.005) units, and the IRR in 1.44 (95%CI 1.15; 1.78, *p* = 0.001) times. However, in the sensitivity analysis (excluding Lima Downtown), a significant association was observed only in the negative binomial mixed-effects models for all AD outcomes. Higher log-Fe concentration was associated with an increase in the IRR of 1.38 (95%CI 1.03; 1.84, *p* = 0.029) times (Table 3).

For the analysis including all districts, when examining the association between log-Fe concentrations in PM_10_ for each separated outcome (AD and DAD), no association was found with the AD rate in the linear (Gaussian) mixed-effects model, but a significant increase in the IRR was observed (1.47 95%CI 1.01; 2.12, *p* = 0.043). For DAD, a statistically significant increase was found for both the rate (6.55 95%CI 1.50; 11.59, *p* = 0.011) and the IRR (1.53 95%CI 1.18; 1.96, *p* = 0.001). In the sensitivity analysis (excluding Lima Downtown), there was only a significant increase in the DAD IRR (1.36 95%CI 1.01; 1.83, *p* = 0.044) (Table 3).

When controlling for other metals (Table 4), a significant association was found between log-Fe concentration and DAD in the Gaussian mixed-effects model, showing an increase of 6.76 (95%CI 0.07; 13.46, *p* = 0.048) in the DAD rate per 100,000 people per log-Fe unit increase. This association was not maintained in the sensitivity analysis model. Regarding the negative binomial mixed-effects analysis, a similar effect size was obtained compared to the model without the other metals. Increased Log-Fe was associated with a higher IRR of AD + DAD (1.55, 95%CI 1.09; 2.20, *p* = 0.014) and with a higher IRR for DAD (1.83, 95%CI 1.21; 2.78, *p* = 0.004). No significant association was observed in the sensitivity analysis. The other metals did not show a significant association, except for Mn, which showed an increase in AD + DAD (IRR = 1.09, 95%CI 1.01; 1.17, *p* = 0.032) and AD in the complete model (IRR = 1.15, 95%CI 1.01; 1.29, *p* = 0.030).

As SO_2_ in air may act synergistically with Fe, its correlation with AD outcomes was evaluated. SO_2_ had a positive correlation with AD cases (r = 0.57; *p* < 0.001), DAD cases (r = 0.34; *p* = 0.001), and AD + DAD cases (r = 0.46; *p* < 0.001) (Table 5). A significant association was found between Fe concentration in PM_10_ with the number of cases and the IRR in the different Alzheimer’s disease outcomes after controlling for SO_2_ (Table 5).

## 4. Discussion

The present study was designed to assess the association between Fe trace element concentrations in PM_10_ and the incidence of AD in six different districts of Lima.

The different trace metal concentrations were lower in Lima compared to those in León (Mexico) [37], but with differences between districts. Comas had the highest iron levels (2096 ng m^−3^). This difference could be partially explained by the level of industrialization of each city; nonetheless, compared with a heavily industrialized area near Athens [38], the Lima districts evaluated showed higher concentrations for Cd, Cu, Mn, Pb, Co, and Fe, even in non-industrial districts such as Santiago de Surco and Lince. These trace metals found in Lima might not only exert an effect on mental health, but also increment the risk of cancer in adults and children as found in other places [37,38,39], and it should be addressed in future studies.

From the 14 metals evaluated in PM_10_, the highest levels of Fe, Mn, Ni, and Pb were observed in Comas, while the lowest levels were in Lince and Santiago de Surco. The highest levels in San Juan de Miraflores were observed for Fe and Cu. These data suggest that there is heterogeneity in the trace elements in PM_10_ in the different districts.

Long-term exposure to PM_10_ has been found to contribute to pathological amyloid-β deposition in adults without dementia [13]. The same has been found for Fe, which not only acts by stimulating amyloid aggregates formed from the β-amyloid peptide [20]; due to its oxidative capacity in the production of ROS, it also contributes to the pathogenicity of AD [21].

Districts in Lima with the lowest Fe concentrations in PM_10_ (Lince and Santiago de Surco) also showed the lowest incidences of AD (alone and with dementia). The literature indicates that AD is more frequently observed at higher ages (≥60 years old) [40] and that air pollution is associated with AD [41,42].

It is possible that the association between increased Fe and AD outcomes could be influenced by the presence of other metals in PM_10_. Nonetheless, Fe was the only metal with a consistent correlation, supporting the idea that Fe trace elements might contribute the most to the progression or severity of AD. On the other hand, Fe had the highest effect size, which was even greater when analyzing Fe alone when controlling the regression models for other metals such as Pb, Cu, Zn, and Mn. Although controlling for the effect of other metals is one approach, future studies should use more sophisticated methods such as Weighted Quantile Sum (WQS) or Bayesian Kernel Machine regression models. These methods can better explore the synergy with other metals, such as manganese, which is associated with cognitive affectation [43].

Zn and Pb have the highest correlation with Fe. The interaction between toxic and essential elements is of particular interest because the deficiency of essential elements can dramatically increase the absorption rate of toxic metals inside the body [44]. Nonetheless, only iron showed a relationship with AD and DAD. However, other studies have evidence of the role of dyshomeostasis of Fe, Cu, and Zn with metal–amyloid interactions that lead to the pathogenesis of AD [45]. Since PM_10_ is composed of several metals, iron might not be the only metal promoting brain damage. Future studies evaluating multi-metal exposure models should be conducted. Further studies will be needed to unveil the nature of these interactions.

No association was observed in the sensitivity analysis model (excluding Lima Downtown) when controlled for other metals. Differences in AD diagnostic capabilities across districts may influence the results [46], making it crucial to standardize case identification to prevent an imbalance in reported cases. Lima Downtown, which benefits from better diagnostic resources through the INCN, exhibited the highest incidence of AD + DAD (69.90 cases per 100,000 inhabitants). This suggests that the overall association observed in the full model may be influenced by the enhanced diagnostic capabilities and case identification in Lima Downtown.

The subjects of this study all lived in each of the six districts that have been established. In the case of Lima Downtown, there is the National Institute of Neurological Sciences, which has among its objectives the dissemination of information about AD as well as detection campaigns.

The association between iron and AD was maintained even after adjusting for SO_2_ exposure. Experimentally, SO_2_ can modulate behavioral effects of Fe inhalation, and brain metal dyshomeostasis may be an important factor in air pollution neurotoxicity [47].

These results align with those of a case–control study in Taiwan, comparing the risk of cognitive impairment in AD with different air pollutants [48]. The mechanism behind the synergy between these two pollutants is suggested to involve SO_2_ facilitating Fe uptake through the bronchial epithelium and altering its intracellular distribution [29]. In the combined model explored in this study, Fe association with AD outcomes was maintained even after controlling for SO_2_. Nonetheless, future studies should thoroughly investigate the interaction and combined effect of SO_2_ and Fe-rich elements in PM_10_. Furthermore, the results including SO_2_ should be taken with caution given the low number of observations and be improved in future studies.

Air pollutants (PM and SO_2_) may also worsen AD symptoms or accelerate the decline of cognitive functions in these patients, as seen in a Korean cohort of 269 patients with mild cognitive impairment or early dementia due to AD, in which PM_10_ was not found to affect the memory capability of the participants, but a 5-year cumulative exposure to SO_2_ showed an association with a decrease in the memory test score employed [49]. This is comparable to the increase in the proportion of AD + dementia in AD, AD alone, and dementia in AD, indicating that Fe exposure may also impact the worsening of AD symptoms or severity.

Exposure to Fe-rich combustion-derived nanoparticles throughout life might promote their accumulation in nervous tissue and structures such as neurons, glia, choroid plexus endothelium, and olfactory epithelium, especially in people exposed to high levels of PM [24]. However, the easy access of Fe to the brain through the olfactory bulb [24,26,27] emphasizes the importance of avoiding air pollution and restricting access for people in zones with high Fe levels in PM. This is a preventable factor.

Data suggest that metals may accumulate in glial cells. Iron has the highest concentration in oligodendrocytes, Cu in astrocytes, and Zn in the glia of the hippocampus and cortex [50]. Cu, Fe, and Mn have neurotoxic effects, while those of Zn can be bidirectional, i.e., neurotoxic but also neuroprotective effects depending on the dose and disease state. Recent data point to the association of metals with neurodegeneration through their role in the modulation of protein aggregation. Metals can accumulate in the brain with aging and may be associated with age-related diseases [50].

The study presents some limitations. Like other countries in Latin America, there might be a significant underdiagnosis of AD cases due to structural barriers [51], which could impact the number of cases in districts without specialized centers like the INCN. These barriers include access to healthcare, fragmented healthcare systems, limited research funding, unstandardized diagnosis and treatment, genetic heterogeneity, and varying social determinants of health. On the other hand, there could be a possible information bias about the district cases, since the data obtained were only aggregated and it was not possible to determine how many people move between districts. Since migration between districts has increased rapidly over the past few decades, consideration should be given to the contributions of mobility.

Taking into consideration that the PM_2.5_ fraction is more bioavailable, future studies should include measurements of iron in PM_2.5_. Another limitation is that important confounders such as comorbidities, socioeconomic status, educational attainment, sex, family history, and clinical conditions could not be adjusted for.

Future studies should also consider a seasonal and spatial distribution assessment of the metals, as well as the influence of other environmental factors such as humidity, on their distribution [52]. The external validity of this study may also be affected since only 6 districts out of the 43 that are part of Lima were considered because active metal monitoring speciation is only done in these districts.

The results of the current study should be interpreted with caution since they are aggregated and analyzed data for geographical areas and not for individuals. Further studies using an individual-based approach should be conducted to evaluate potential confounders and avoid the ecological fallacy (to assume that all participants have the same exposure level), as exposure levels can be highly heterogeneous among patients. However, the high pollution in cities like Lima, Peru, and their association with morbidity and mortality, as described previously [53,54,55,56], reveals that regulations to maintain low PM emissions are not observed in the country.

The present study shows us that the simultaneous presence of trace metals in particulate matter can generate synergies and antagonisms that may have implications in the pathology studied. In many of these metals in the PM, there are no recommended reference values, or in other cases the values that are established may not be optimal because, due to the synergies, it is possible that negative effects can develop with lower values of the metal under study, even at doses recommended by the regulatory bodies.

## 5. Conclusions

The presence of elevated Fe trace elements in PM_10_ resulting from air pollution in various districts in Lima was found to be correlated with a higher incidence of AD and dementia in AD cases in the ≥60-year-old population, compared to districts with lower Fe levels in PM_10_. These findings underscore the potential risk of exposure to heavy metal pollutants in neurodegenerative diseases, including AD, emphasizing the need for the development of air pollution control guidelines in Peru.

## Figures and Tables

**Figure 1 biomedicines-12-02043-f001:**
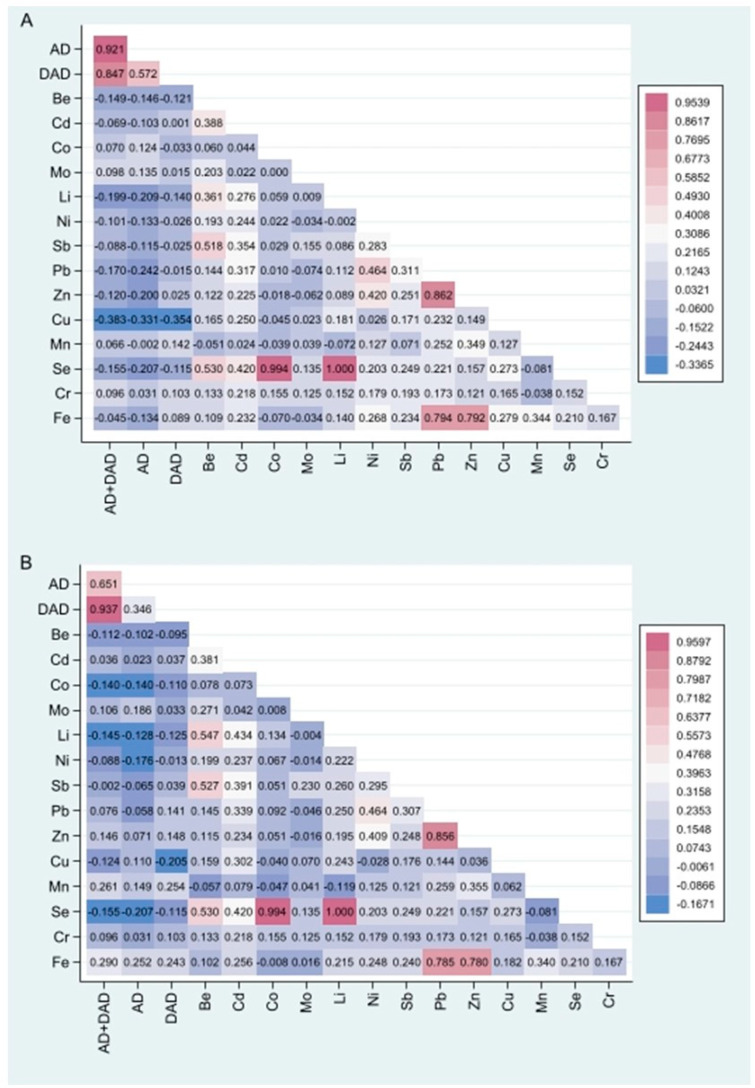
Pearson correlation analysis of the different trace metals in PM_10_ and Alzheimer’s disease (AD), dementia in Alzheimer’s disease (DAD), and AD + DAD including all the districts (**A**) and excluding Lima Downtown (**B**). Pearson coefficient (r) value is inside colored boxes. Colors closer to red mean a more positive correlation, while blue shades indicate a less positive correlation, and a color closer to solid blue means a more negative correlation. For (**A**), the correlation of AD + DAD with Li, Pb, Zn, Be and Cu was statistically significant (*p* < 0.05). The correlation of AD with Ni, Se, Mo, and Fe was statistically significant (*p* < 0.05). The correlation of DAD with Fe, Mn, Li, and Cu was statistically significant (*p* < 0.05). The correlation of Fe with Cd, Li, Ni, Sb, Pb, Zn, Cu, Mn, Cr, and Se was statistically significant (*p* < 0.05). For (**B**), the correlation of AD + DAD with Li, Se, Mn, and Fe was statistically significant (*p* < 0.05). The correlation of AD with Mo and Fe was statistically significant (*p* < 0.05). The correlation of DAD with Fe, Mn, and Cu was statistically significant (*p* < 0.05). The correlation of Fe with Cd, Li, Ni, Sb, Pb, Zn, Cu, Se, Cr, and Mn was statistically significant (*p* < 0.05).

**Figure 2 biomedicines-12-02043-f002:**
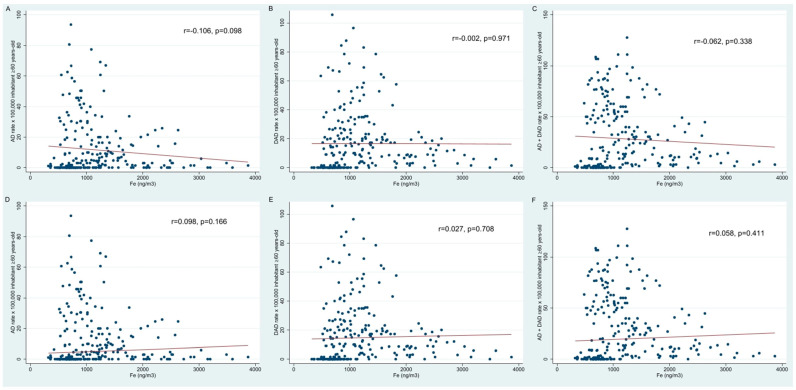
Scatter plots between Alzheimer’s disease (AD) cases (**A**), dementia in Alzheimer’s disease (DAD) cases (**B**), and DAD + AD cases (**C**) and iron (ng/m^3^) in PM_10_ considering all the districts, and excluding Lima Downtown data ((**D**–**F**) respectively). The red line represents a perfect linear relationship between the outcomes and Fe concentration.

**Table 1 biomedicines-12-02043-t001:** The concentration of 14 metals (ng/m^3^) present PM_10_ in six districts of Metropolitan Lima: Comas, Lima Downtown, Lince, El Agustino, San Juan de Miraflores, and Santiago de Surco.

Metals (ng/m^3^ ± SD)	Comas	Lima Downtown	Lince	El Agustino	San Juan de Miraflores	Santiago de Surco	*p*-Value ^¥^
#Observations	41	41	42	40	38	40	
Fe	2096 ± 750	868 ± 223	692 ± 172	1089 ± 319	1568 ± 507	694 ± 178	>0.001
Cd	2.4 ± 0.3	2.3 ± 0.2	2.3 ± 0.2	2.3 ± 0.7	2.3 ± 0.0	2.3 ± 0.1	0.174
Cr	6.0 ± 1.4	5.6 ± 0.5	5.5 ± 0.7	6.1 ± 2.9	5.8 ± 0.8	6.0 ± 3.4	0.180
Cu	63.1 ± 31.3	34.6 ± 12.5	44.8 ± 17.7	43.4 ± 53.9	72.0 ± 27.1	56.3 ± 21.9	>0.001
Li	168.7 ± 2.9	166.9 ± 4.3	165.6 ± 5.9	165.2 ± 8.6	167.1 ± 1.8	167.7 ± 5.2	0.014
Mn	44.3 ± 26.4	17.0 ± 12.9	14.0 ± 8.0	24.9 ± 16.3	25.3 ± 16.0	14.6 ± 11.0	>0.001
Mo	4.2 ± 0.4	4.2 ± 0.6	4.4 ± 2.6	4.6 ± 3.7	4.3 ± 1.3	4.1 ± 0.1	0.660
Ni	8.1 ± 3.1	6.8 ± 1.5	6.5 ± 1.0	6.7 ± 2.3	6.5 ± 0.6	7.0 ± 2.6	>0.001
Pb	53.9 ± 29.3	18.5 ± 7.4	15.5 ± 3.5	25.5 ± 19.1	21.6 ± 8.8	15.8 ± 6.1	>0.001
Sb	12.2 ± 3.1	11.6 ± 1.7	11.6 ± 2.8	12.1 ± 4.2	11.5 ± 0.7	11.4 ± 0.4	0.296
Se	71.7 ± 1.2	70.9 ± 1.8	70.4 ± 2.5	70.2 ± 3.6	71.0 ± 0.8	71.3 ± 2.2	0.526
Zn	300.9 ± 119.5	95.5 ± 42.2	69.1 ± 22.3	118.4 ± 49.6	139.6 ± 81.8	75.9 ± 26.7	>0.001
Be	0.9 ± 0.01	0.9 ± 0.02	0.9 ± 0.1	0.9 ± 0.05	0.9 ± 0.001	0.9 ± 0.02	0.726
Co	6.8 ± 0.1	6.8 ± 0.1	6.7 ± 0.2	6.7 ± 0.2	6.6 ± 1.1	6.8 ± 0.2	0.463

^¥^ One-way ANOVA.

**Table 2 biomedicines-12-02043-t002:** AD + dementia in AD (G30 + F00), AD alone (G30), dementia in AD alone (F00) cases and incidences by district.

District	Comas	Lima Downtown	Lince	El Agustino	San Juan de Miraflores	Santiago de Surco	*p*-Value
Number of observations	41	41	42	40	38	40	
AD + DAD (%)	201 (7.50)	1373 (51.25)	22 (0.82)	581 (21.69)	478 (17.84)	24 (0.90)	<0.001
AD cases (%)	34 (2.82)	857 (71.18)	17 (1.41)	92 (7.65)	191 (15.86)	13 (1.08)	<0.001
DAD cases (%)	167 (11.32)	516 (34.98)	5 (0.34)	489 (33.15)	287 (19.46)	11 (0.75)	<0.001
AD + DAD incidence (mean ± SD) ^α^	6.83 ± 4.49	69.90 ± 22.78	4.50 ± 14.75	62.28 ± 23.17	26.72 ± 9.86	0.89 ± 1.06	<0.001 ^£^
AD incidence (mean ± SD) ^α^	1.17 ± 1.47	43.01 ± 16.07	3.44 ± 14.72	10.83 ± 9.76	10.86 ± 7.01	0.44 ± 0.75	<0.001 ^£^
DAD incidence (mean ± SD) ^α^	5.66 ± 3.91	25.88 ± 11.45	1.06 ± 2.91	51.44 ± 24.07	15.86 ± 5.53	0.45 ± 0.91	<0.001 ^£^

AD: Alzheimer’s disease. DAD: dementia in Alzheimer’s disease. G30 + F00 (AD + DAD), G30 (AD), and F00 (DAD) total counts of cases for the study years are presented. ^α^ Incidence per 100,000 people for ≥60-year-old population. Percentages were compared by chi-square test. ^£^ One-way ANOVA test.

**Table 3 biomedicines-12-02043-t003:** Analysis of log-iron (log-Fe) concentration with Alzheimer’s disease and dementia in Alzheimer’s disease.

Outcome	Mixed Effects	Complete Model β-Coeff (CI 95%)	Sensitivity Analysis Model IRR (CI 95%)
AD + DAD	Gaussian	9.93 (2.93; 16.93) *	5.77 (−0.97; 12.51)
	Negative binomial	1.44 (1.15; 1.78) *	1.38 (1.03; 1.84) *
AD	Gaussian	3.29 (−1.50; 8.08)	1.65 (−2.35; 5.64)
	Negative binomial	1.47 (1.01; 2.12) *	1.35 (0.76; 2.38)
DAD	Gaussian	6.55 (1.50; 11.59) *	4.31 (−1.27; 9.90)
	Negative binomial	1.53 (1.18; 1.96) *	1.36 (1.01; 1.83) *

Linear mixed-effects (Gaussian) and mixed-effects negative binomial (Negative binomial) hierarchical models were constructed. Complete model includes all study districts, the population at risk considered was ≥60 years of age. Negative binomial mixed-effects model used ≥60-year-old population as the offset. Sensitivity analysis model excludes Lima Downtown. Models were adjusted for month and year. IRR: incidence rate ratio. CI: confidence interval at 95%. AD: Alzheimer’s disease. DAD: dementia in Alzheimer’s disease. All Wald tests comparing the mixed-effects model with a regular linear model had *p* < 0.05. * Statistical significance *p* < 0.05.

**Table 4 biomedicines-12-02043-t004:** Multivariate regression analysis between the log concentrations of Fe, Pb, Cu, Zn, and Mn with AD + DAD, AD, and DAD cases.

Outcome/Metal		Gaussian Mixed Effects		Negative Binomial Mixed Effects	
		Complete Model	Sensitivity Analysis Model	Complete Model	Sensitivity Analysis Model
AD + DAD					
	log-Fe	8.91 (−0.83; 18.66)	4.47 (−4.46; 13.40)	1.55 (1.09; 2.20) *	1.43 (0.93; 2.21)
	log-Pb	−7.11 (−16.09; 1.87)	−5.45 (−13.58; 2.67)	0.73 (0.53; 1.01)	0.73 (0.50; 1.07)
	log-Cu	0.49 (−5.18; 6.15)	−0.58 (−5.65; 4.49)	0.92 (0.77; 1.09)	0.90 (0.74; 1.10)
	log-Zn	1.04 (−7.55; 9.62)	0.26 (−7.67; 8.19)	1.00 (0.75; 1.32)	0.95 (0.67; 1.33)
	log-Mn	2.03 (−0.33; 4.39)	1.36 (−0.83; 3.54)	1.09 (1.01; 1.17) *	1.09 (0.99; 1.18)
AD					
	log-Fe	2.14 (−4.83; 9.12)	1.62 (−4.50; 7.74)	1.32 (0.74; 2.34)	1.51 (0.62; 3.71)
	log-Pb	−3.09 (−9.52; 3.34)	−1.76 (−7.37; 3.85)	0.72 (0.43; 1.22)	0.77 (0.35; 1.69)
	log-Cu	1.65 (−2.41; 5.71)	0.31 (−3.24; 3.86)	0.99 (0.43; 1.22)	0.88 (0.59; 1.32)
	log-Zn	0.15 (−6.00; 6.30)	−0.38 (−5.85; 5.09)	1.07 (0.68; 1.68)	0.84 (0.42; 1.67)
	log-Mn	1.37 (−0.32; 3.06)	0.73 (−0.81; 2.28)	1.15 (1.01; 1.29) *	1.11 (0.93; 1.34)
DAD					
	log-Fe	6.76 (0.07; 13.46) *	3.86 (−3.06; 10.79)	1.83 (1.21; 2.78) **	1.51 (0.96; 2.37)
	log-Pb	−4.03 (−10.20; 2.14)	−4.11 (−10.42; 2.19)	0.75 (0.52; 1.08)	0.70 (0.47; 1.05)
	log-Cu	−1.27 (−5.17; 2.62)	−0.97 (−4.90; 2.96)	0.88 (0.71; 1.08)	0.92 (0.74; 1.13)
	log-Zn	0.81 (−5.09; 6.71)	0.42 (−5.73; 6.56)	0.96 (0.69; 1.34)	0.94 (0.66; 1.35)
	log-Mn	0.67 (−0.95; 2.30)	0.62 (−1.08; 2.31)	1.04 (0.95; 1.14)	1.06 (0.96; 1.16)

Linear mixed-effects (Gaussian) and mixed-effects negative binomial (Negative binomial) hierarchical models were constructed. Complete model includes all study districts, the population at risk considered was ≥60 years of age. Negative binomial GLM used ≥60-year-old population as the offset. Sensitivity analysis model excludes Lima Downtown. Gaussian mixed-effects shows an adjusted β-coefficient (95% confidence interval). Negative binomial mixed-effects model shows an adjusted incidence rate ratio (95% confidence interval). Models were adjusted for the log-concentrations of Pb, Cu, Zn, and Mn, and for month and year. AD + DAD: Alzheimer’s disease (AD) and dementia in Alzheimer’s disease (DAD) combined cases. All Wald tests comparing the mixed-effects model with a regular linear model had *p* < 0.05. * *p* < 0.05, ** *p* < 0.01.

**Table 5 biomedicines-12-02043-t005:** Correlations between the cases (AD, dementia in AD, and AD + dementia in AD) with SO_2_, and generalized linear models between total AD cases (dementia in AD + AD) and log-Fe and SO_2_.

Analysis/Outcome		GLM Gaussian	GLM Negative Binomial ^#^	Correlation ^#^
AD + DAD				
	Fe	1.21 (0.72–1.70) **	7.06 (4.26–11.68) **	-
	SO_2_	0.04 (0.02–0.05) **	1.04 (1.02–1.05) **	0.57 **
AD				
	Fe	0.52 (0.23; 0.80) **	9.42 (4.55; 19.48) **	-
	SO_2_	0.02 (0.01; 0.03) **	1.05 (1.04; 1.07) **	0.34 **
DAD				
	Fe	0.70 (0.43; 0.97) **	5.90 (3.50; 9.95) **	-
	SO_2_	0.02 (0.01; 0.03) **	1.03 (1.02; 1.04) **	0.46 **

**^#^** Pearson correlation analysis. Only the Pearson r correlation coefficient for SO_2_ is shown since the correlation coefficient for log-Fe with the different outcomes can be found in Figure 1. GLM: generalized linear model. β-coefficient (95% confidence interval) is shown. Negative binomial GLM considered the ≥60-year-old district population as the offset. Incidence rate ratio (95% confidence interval) is shown. AD: Alzheimer’s disease. DAD: dementia in Alzheimer’s disease. Fe: iron; SO_2_: sulfur dioxide. **** Statistical significance *p* < 0.01.

## Data Availability

The original data presented in the study are openly available in a Zenodo repository (https://zenodo.org/) at https://doi.org/10.5281/zenodo.11640086 (accessed on 3 March 2023).

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
