# Peer review of "Iron Trace Elements Concentration in PM10 and Alzheimer’s Disease in Lima, Peru: Ecological Study"

_biomedicines, 2024, doi:10.3390/biomedicines12092043_

Round 1

Reviewer 1 Report (Previous Reviewer 1)

Comments and Suggestions for Authors

The authors have addressed the previous comments.

Author Response

Comment 1: The authors have addressed the previous comments.

Response: Dear reviewer, thank you for your appreciation

Reviewer 2 Report (Previous Reviewer 2)

Comments and Suggestions for Authors

This paper is a resubmission of a previous manuscript and it has improved from the previous version. However, we notice the lack of a brief review about the present state of the art on the chemical factors that may be responsible for the AD and the role of the present paper in this state of the art. 

Author Response

Comment 1: This paper is a resubmission of a previous manuscript, and it has improved from the previous version. However, we notice the lack of a brief review about the present state of the art on the chemical factors that may be responsible for the AD and the role of the present paper in this state of the art. 

ANSWER:

Dear reviewer, thank you for your comment. We have included several references for which state of the art about metals linked to AD. We have added the next text:

“Neurodegeneration occurs due to the association between metals and proteins, which is then followed by aggregate formation, mitochondrial disorder, and, ultimately, cell death. High levels of iron (Fe), copper (Cu), and manganese are associated with mild cognitive impairment (MCI) and Alzheimer's disease (AD), while low selenium (Se) levels are linked to poor cognitive status. In Alzheimer's disease, low Zn levels suppress the neurotoxicity induced by β-amyloid through the selective precipitation of aggregation intermediates. In AD, at low levels, Zn suppresses β-amyloid-induced neurotoxicity by selectively precipitating aggregation intermediates; however, at high levels, the binding of Zn to β-amyloid may enhance the formation of fibrillar β-amyloid aggregation, leading to neurodegeneration. (TyczyÅ„ska et al., 2024; Mateo et al., 2023).

High levels of Cu, Mn, and Fe participate in the formation of α-synuclein aggregates in intracellular inclusions, called Lewy Body, that result in synaptic dysfunction and interruption of axonal transport. In AD a diffuse accumulation of Fe occurs in various regions, such as cortex and hippocampus, with Fe marginally increased in the senile plaques (Mezzaroba et al., 2019) . (TyczyÅ„ska et al., 2024; Mateo et al., 2023).

High aluminum (Al) exposure is associated with frontotemporal dementia (FTD), and elevated selenium (Se) levels may be linked to its onset. Aluminum disturbs the homeostasis of other metals through a rise in the production of oxygen-reactive forms, which then leads to cellular death. Selenium, in association with iron, plays a distinct role in the process of ferroptosis (Tyczyńska et al., 2024; Mateo et al., 2023).

  • Mezzaroba L, Alfieri DF, Colado Simão AN, Vissoci Reiche EM. The role of zinc, copper, manganese, and iron in neurodegenerative diseases. Neurotoxicology. 2019 Sep;74:230-241. doi: 10.1016/j.neuro.2019.07.007. Epub 2019 Aug 1. PMID: 31377220.
  • Mateo D, Marquès M, Torrente M. Metals linked with the most prevalent primary neurodegenerative dementias in the elderly: A narrative review. Environ Res. 2023 Nov 1;236(Pt 1):116722. doi: 10.1016/j.envres.2023.116722. Epub 2023 Jul 23. PMID: 37487923.
  • TyczyÅ„ska M, GÄ™dek M, Brachet A, StrÄ™k W, Flieger J, TeresiÅ„ski G, Baj J. Trace Elements in Alzheimer's Disease and Dementia: The Current State of Knowledge. J Clin Med. 2024 Apr 19;13(8):2381. doi: 10.3390/jcm13082381. PMID: 38673657; PMCID: PMC11050856.

Trace elements such as iron (Fe), zinc (Zn), copper (Cu), and manganese (Mn) are absorbed from food via the gastrointestinal tract, transported into the brain, and play central roles in normal brain functions. An excess of these trace elements often produces reactive oxygen species and damages the brain. Moreover, increasing evidence suggests that the dyshomeostasis of these metals is involved in the pathogenesis of neurodegenerative diseases, including Alzheimer's disease, prion diseases, and Lewy body diseases (Kawahara et al., 2023). 

  • Kawahara M, Kato-Negishi M, Tanaka KI. Dietary Trace Elements and the Pathogenesis of Neurodegenerative Diseases. Nutrients. 2023 Apr 25;15(9):2067. doi: 10.3390/nu15092067. PMID: 37432185; PMCID: PMC10180548.
  • This paper adds information that iron in particulate matter (PM10) is also a source of high iron in the brain and a risk for AD.
  • Oxidative stress resulting from abnormal homeostasis of transition metals such as iron, copper, and zinc may be a causative explanation behind AD. In the nervous system, the interaction of metals with proteins are essential variable in the development or suppression of neurodegeneration. Metal buildup in the nervous system, as reported in the AD, could be the result of compensatory mechanisms designed to improve metal availability for physiological functions (Singh et al., 2024).

Singh R, Panghal A, Jadhav K, Thakur A, Verma RK, Singh C, Goyal M, Kumar J, Namdeo AG. Recent Advances in Targeting Transition Metals (Copper, Iron, and Zinc) in Alzheimer's Disease. Mol Neurobiol. 2024 May 29. doi: 10.1007/s12035-024-04256-8. Epub ahead of print. PMID: 38809370.

Reviewer 3 Report (New Reviewer)

Comments and Suggestions for Authors

The authors examine the coherence between Fe concentration and of other metals in the air and the occurrence of AD in elder people in Lima in Peru. The coherence between the concentration of these metals and the formation of AD should be described more exactly. Besides, the importanc of Al should be mentioned, although it has not been examined. The English language should be controlled. It is an important article.

Comments on the Quality of English Language

The English language is sufficiently good, it should be controlled.

Author Response

Comment 1: The authors examine the coherence between Fe concentration and other metals in the air and the occurrence of AD in elderly people in Lima Peru. The coherence between the concentration of these metals and the formation of AD should be described more exactly. Besides, the importance of Al should be mentioned, although it has not been examined. The English language should be controlled. It is an important article.

ANSWER:

Dear reviewer, thank you for the feedback. We have added the next text: “Neurodegeneration occurs due to the association between metals and proteins, which is then followed by aggregate formation, mitochondrial disorder, and, ultimately, cell death. High levels of iron (Fe), copper (Cu), and manganese are associated with mild cognitive impairment (MCI) and Alzheimer's disease (AD), while low selenium (Se) levels are linked to poor cognitive status. In Alzheimer's disease, low Zn levels suppress the neurotoxicity induced by β-amyloid through the selective precipitation of aggregation intermediates. In AD, at low levels, Zn suppresses β-amyloid-induced neurotoxicity by selectively precipitating aggregation intermediates; however, at high levels, the binding of Zn to β-amyloid may enhance the formation of fibrillar β-amyloid aggregation, leading to neurodegeneration (TyczyÅ„ska et al., 2024; Mateo et al., 2023).

  • High levels of Cu, Mn, and Fe participate in the formation of α-synuclein aggregates in intracellular inclusions, called Lewy Body, that result in synaptic dysfunction and interruption of axonal transport. In AD a diffuse accumulation of Fe occurs in various regions, such as the cortex and hippocampus, with Fe marginally increased in the senile plaques (Mezzaroba et al., 2019; TyczyÅ„ska et al., 2024; Mateo et al., 2023).
  • High aluminum (Al) exposure is associated with frontotemporal dementia (FTD), and elevated selenium (Se) levels may be linked to its onset. Aluminum disturbs the homeostasis of other metals through a rise in the production of oxygen-reactive forms, which then leads to cellular death. Selenium, in association with iron, plays a distinct role in the process of ferroptosis (TyczyÅ„ska et al., 2024; Mateo et al., 2023)”.
  • Mezzaroba L, Alfieri DF, Colado Simão AN, Vissoci Reiche EM. The role of zinc, copper, manganese, and iron in neurodegenerative diseases. Neurotoxicology. 2019 Sep;74:230-241. doi: 10.1016/j.neuro.2019.07.007. Epub 2019 Aug 1. PMID: 31377220.
  • Mateo D, Marquès M, Torrente M. Metals linked with the most prevalent primary neurodegenerative dementias in the elderly: A narrative review. Environ Res. 2023 Nov 1;236(Pt 1):116722. doi: 10.1016/j.envres.2023.116722. Epub 2023 Jul 23. PMID: 37487923.
  • TyczyÅ„ska M, GÄ™dek M, Brachet A, StrÄ™k W, Flieger J, TeresiÅ„ski G, Baj J. Trace Elements in Alzheimer's Disease and Dementia: The Current State of Knowledge. J Clin Med. 2024 Apr 19;13(8):2381. doi: 10.3390/jcm13082381. PMID: 38673657; PMCID: PMC11050856.
  • “Trace elements such as iron (Fe), zinc (Zn), copper (Cu), and manganese (Mn) are absorbed from food via the gastrointestinal tract, transported into the brain, and play central roles in normal brain functions. An excess of these trace elements often produces reactive oxygen species and damages the brain. Moreover, increasing evidence suggests that the dyshomeostasis of these metals is involved in the pathogenesis of neurodegenerative diseases, including Alzheimer's disease, prion diseases, and Lewy body diseases (Kawahara et al., 2023)”. 
  • Kawahara M, Kato-Negishi M, Tanaka KI. Dietary Trace Elements and the Pathogenesis of Neurodegenerative Diseases. Nutrients. 2023 Apr 25;15(9):2067. doi: 10.3390/nu15092067. PMID: 37432185; PMCID: PMC10180548.

“This paper adds information that iron in particulate matter (PM10) is also a source for high iron in the brain and a risk for AD”.

Comments on the Quality of English Language

The English language is sufficiently good, it should be controlled.

Reviewer 4 Report (New Reviewer)

Comments and Suggestions for Authors

The manuscript is written clearly and understandably. It is interesting to read. I think that a number of points in the manuscript need improvement. I will try to help the authors to improve the manuscript.
1.The introduction is biased and a number of facts are presented in a very one-sided way. The authors repeatedly throw in the idea that there is a proven, obvious link between the concentration of nanoparticles in the air and the likelihood of developing Alzheimer's disease. The authors probably remember how in the 1990s there were attempts to link the concentration of heavy and transition metal ions to the likelihood of developing the disease. Later, the development of Alzheimer's disease was linked to the presence of aluminum in food. Even in some countries there was hysteria about the use of aluminum cookware. Then there was a trend to research various organometallic compounds. Now nanoparticles... Perhaps the authors should write an introduction with this information in mind. We do not want to develop hysteria in society about the use of iron cutlery or the mining of magnetite, hematite or limonite, do we?

2. Perhaps the authors need to state their position on the cause of Alzheimer's disease! Write that the cause of Alzheimer's disease is not completely clear. That most likely the cause is not a single chemical compound, but a complex of external influences and internal changes occurring in the body. That there are several hypotheses of the development of Alzheimer's disease and the data used to build these hypotheses in some way correlate with the accumulation in some parts of the brain of metals, including iron.

3.When the authors write about PM10, they refer to the work on PM2.5. Perhaps we need to explain this change in context.

4. In my opinion, the heart of the work is statistical analysis. It seems to me that the use of certain statistical approaches should be justified in more detail.It is not always clear why the authors took this or that path. For example, when using the Bonferroni test, how did the authors protect themselves from "Bonferroni chaos"? How do the authors assess the validity of such data?It may be necessary to write very briefly, in the text of the manuscript, the reasons for choosing certain analysis tools.

5. The authors write: "Correlations between Fe and other metals present in the PM10 samples are provided in Supplementary Material 1.". Unfortunately, the authors forgot to Supplementary Material 1. Please do so.

6. The discussion section of the manuscript is not related to the results obtained. I did not find in this section any reference to the results obtained in the manuscript. The authors in the discussion section once again analyze the literature... I think this situation should be corrected. In the discussion section everyone expects to see an answer to the question, what follows from the results obtained?How do the results help to interpret the data obtained earlier?What new things have we learned? I urge the authors to revise the discussion section of the manuscript.

7. What new things did we learn from the Conclusion of the manuscript? That the city of Lima also has a correlation between the likelihood of developing Alzheimer's disease and air pollution? That we need to control air pollution in metropolitan areas? That it is important to control the concentration of SO4 in the atmosphere? I think the authors need to rework the conclusion somewhat.

Minor point
- ".2. Materials and Methods" - need to remove the period before the chapter title
- There is no clearly labeled title for Figures 1A and 1B. Instead of a title, examples of the correlations in the figures are given.

Author Response

Dear reviewer, we appreciate your feedback. In the enclosed file, please find our responses.

Round 2

Reviewer 2 Report (Previous Reviewer 2)

Comments and Suggestions for Authors

The authors have answered satisfactorily to the referee suggestions. 

Author Response

Dear Reviewer,

We appreciate your comments.

Bests,

Reviewer 4 Report (New Reviewer)

Comments and Suggestions for Authors

I really hope that I helped the authors make their manuscript better.

Author Response

Dear Reviewer,

We appreciate your comments, they were very helpful.

Bests,

This manuscript is a resubmission of an earlier submission. The following is a list of the peer review reports and author responses from that submission.

Round 1

Reviewer 1 Report

Comments and Suggestions for Authors

This extremely brief manuscript looks for a relationship between previously available data on PM and its iron content and on Alzheimer's disease in Lima, Peru.  While brief, it is interesting, generally well written, and appears to be properly performed.

Care should be taken in table 1 with the significant figures.  If a number has three significant figure; the percentage it represents should also be reported to three significant figures, etc.

Comments on the Quality of English Language

While the punctuation is very good, numerous minor items were found.  IN line two of the abstract and else where "PM10 is comprised of..." not "PM comprised". The third sentence of the abstract should start with "This".  IN the last sentence of the abstract, "concentrations" cannot be "regulated" in "monitoring".

Avoid use of first person voice (e.g., "we" and "our") outside of introduction and conclusion. 

SO2 should be SO2 throughout.

Author Response

Dear Reviewer, thank you for the exhaustive review of the manuscript, the authors have proceeded to carry out a general revision of the manuscript, and the following specific observations are being addressed:

Comment 1: This extremely brief manuscript looks for a relationship between previously available data on PM and its iron content and on Alzheimer's disease in Lima, Peru.  While brief, it is interesting, generally well written, and appears to be properly performed.

Care should be taken in table 1 with the significant figures.  If a number has three significant figures; the percentage it represents should also be reported to three significant figures, etc.

Answer: Table 1 has been corrected, thank you.  

Comment 2: While the punctuation is very good, numerous minor items were found.  IN line two of the abstract and elsewhere "PM10 is comprised of..." not "PM comprised". The third sentence of the abstract should start with "This".  IN the last sentence of the abstract, "concentrations" cannot be "regulated" in "monitoring".

Answer: All these changes has been done in the revised version, thank you.

Comment 3: Avoid use of first-person voice (e.g., "we" and "our") outside of introduction and conclusion.

Answer: Corrections have been made, thank you. We have deleted all the mentions on first-person voice.

Comment 4: SO2 should be SO2 throughout.

Answer: Corrections have been made, thank you.

Reviewer 2 Report

Comments and Suggestions for Authors

This paper describes a study about the correlation of iron presence in PM10 and Alzheimer's disease. I believe that the authors oversimplified the comprehension of such a complex problem that is Alzheimer's disease. Further information should be provided about the samples under analysis:

- The full chemical composition of the PM10 under analysis should be given.

- Taking into consideration that the PM2,5 fraction is more bioavailable the research should focus on this fraction.

- Information about Iron bioavailability in the PM should be obtained. 

Author Response

Dear Reviewer, thank you for the exhaustive review of the manuscript, the authors have proceeded to carry out a general revision of the manuscript, and the following specific observations are being addressed:

Comment 1: This paper describes a study about the correlation of iron presence in PM10 and Alzheimer's disease. I believe that the authors oversimplified the comprehension of such a complex problem that is Alzheimer's disease. Further information should be provided about the samples under analysis: The full chemical composition of the PM10 under analysis should be given.

Answer: The methodology employed for the PM10 speciation and SO2 quantification has been included in the Materials and Methods section of the manuscript.

Ambient air PM10 samples were collected on quartz filters on a weekly basis. These samples underwent chemical analysis using inductively coupled plasma-mass spectrometry (ICP-MS) [28] to determine the concentrations of various heavy metals. DIGESA performs an extensive speciation process of 20 metals.

The Fluorescence SO2 Analyzer operates on the principle that SO2 molecules absorb ultraviolet light. Upon absorbing this light, the SO2 molecules become excited and subsequently emit light as they return to their ground state. The intensity of the emitted light is measured, and this intensity is directly proportional to the concentration of SO2 in the air.

  1. Jamwal R, Manisha Sengar M, Jamwal R. Determination of geographical origin of Mustard oil based on multi-elemental finger-printing using inductively coupled plasma mass spectrometry (ICP-MS) and chemometric analysis. Food Chemistry Advances. 2023; 2, 100233. doi: 10.1016/j.focha.2023.100233.

Comment 2: Taking into consideration that the PM2,5 fraction is more bioavailable the research should focus on this fraction.

Answer: There is evidence that long-term exposure to PM10 has been found to contribute to pathological amyloid-β deposition in adults without dementia [36]. The same has been found for iron, which acts by stimulating amyloid aggregates formed from the β-amyloid peptide [37] and, increasing the production of reactive oxygen species (ROS), contributing to the pathogenicity of AD [38]. PM10 is composed of several other metals, meaning that iron might not be the only metal promoting brain damage. Future studies evaluating multi-metal exposure models should be conducted.

In regard of the influence of PM on Alzheimer’s disease (AD), many human studies demonstrated that PM2.5 was associated with increased risk of AD dementia or cognitive impairment.  Although less studied PM10 was also reported to be related with AD dementia and amnestic type mild cognitive impairment, a high-risk state of AD dementia, and beta amyloid deposition [36]. We have included a statement that future studies should include measurements of iron in PM2.5: “Taking into consideration that the PM2,5 fraction is more bioavailable future studies should include measurements of iron in PM2.5

  1. Lee JH, Byun MS, Yi D, Ko K, Jeon SY, Sohn BK, et al. Long-Term Exposure to PM10 and in vivo Alzheimer's Disease Pathologies. J Alzheimers Dis. 2020;78(2):745-56.
  2. Islam F, Shohag S, Akhter S, Islam MR, Sultana S, Mitra S, et al. Exposure of metal toxicity in Alzheimer's disease: An extensive review. Front Pharmacol. 2022;13:903099.
  3. Opazo C. Metales de transición y enfermedad de Alzheimer. Revista Española de Geriatría y Gerontología. 2005;40(6):365-70.

In Peru, data on speciation is generated in PM10 and as this study was bases from data obtained from the government offices, no available data on iron in PM2.5 exist. This was also included as limitation.

Comment 3: Information about Iron bioavailability in the PM should be obtained.

Answer: We are including it in the discussion section, thank you.

To examine associations of long-term exposure of total and source-specific PM2.5 with incident dementia in older adults in a nationally representative, population-based cohort study in the US.  In single pollutant models, PM2.5 from all sources, except dust, were associated with increased rates of dementia, with the strongest associations for agriculture, traffic, coal combustion, and wildfires. These data provide further evidence supporting PM2.5 reduction as a population-based approach to promote healthy cognitive aging. These findings also indicate that intervening on key emission sources might have value (Zhang et al., 2023).

Zhang B, Weuve J, Langa KM, D'Souza J, Szpiro A, Faul J, Mendes de Leon C, Gao J, Kaufman JD, Sheppard L, Lee J, Kobayashi LC, Hirth R, Adar SD. Comparison of Particulate Air Pollution From Different Emission Sources and Incident Dementia in the US. JAMA Intern Med. 2023 Oct 1;183(10):1080-1089. doi: 10.1001/jamainternmed.2023.3300. PMID: 37578757; PMCID: PMC10425875.

Round 2

Reviewer 2 Report

Comments and Suggestions for Authors

The authors have somewhat improved the manuscript.